# Maximizing Anticancer Response with MPS1 and CENPE Inhibition Alongside Apoptosis Induction

**DOI:** 10.3390/pharmaceutics16010056

**Published:** 2023-12-29

**Authors:** Bárbara Pinto, João P. N. Silva, Patrícia M. A. Silva, Daniel José Barbosa, Bruno Sarmento, Juliana Carvalho Tavares, Hassan Bousbaa

**Affiliations:** 1UNIPRO—Oral Pathology and Rehabilitation Research Unit, University Institute of Health Sciences (IUCS), Cooperativa de Ensino Superior Politécnico e Universitário (CESPU), Rua Central de Gandra, 1317, 4585-116 Gandra, Portugal; barbara.fernandes.15@gmail.com (B.P.); a30242@alunos.cespu.pt (J.P.N.S.); 2Department of Physiology and Biophysics, Institute of Biological Sciences, Federal University of Minas Gerais (UFMG), Av. Pres. Antônio Carlos, 6627, Belo Horizonte 31270-901, Brazil; julianact@ufmg.br; 31H-TOXRUN—One Health Toxicology Research Unit, University Institute of Health Sciences, CESPU, CRL, 4585-116 Gandra, Portugal; daniel.barbosa@iucs.cespu.pt; 4i3S—Institute for Research and Innovation in Health, University of Porto, Rua Alfredo Allen 208, 4200-135 Porto, Portugal; bruno.sarmento@i3s.up.pt; 5INEB—Institute of Biomedical Engineering, University of Porto, Rua Alfredo Allen 208, 4200-393 Porto, Portugal

**Keywords:** CENPE inhibitor, MPS1 inhibitor, BCL-2 family inhibitor, antimitotics, antitumoral activity, combination therapy, cancer treatment

## Abstract

Antimitotic compounds, targeting key spindle assembly checkpoint (SAC) components (e.g., MPS1, Aurora kinase B, PLK1, KLP1, CENPE), are potential alternatives to microtubule-targeting antimitotic agents (e.g., paclitaxel) to circumvent resistance and side effects associated with their use. They can be classified into mitotic blockers, causing SAC-induced mitotic arrest, or mitotic drivers, pushing cells through aberrant mitosis by overriding SAC. These drugs, although advancing to clinical trials, exhibit unsatisfactory cancer treatment outcomes as monotherapy, probably due to variable cell fate responses driven by cyclin B degradation and apoptosis signal accumulation networks. We investigated the impact of inhibiting anti-apoptotic signals with the BH3-mimetic navitoclax in lung cancer cells treated with the selective CENPE inhibitor GSK923295 (mitotic blocker) or the MPS1 inhibitor BAY1217389 (mitotic driver). Our aim was to steer treated cancer cells towards cell death. BH3-mimetics, in combination with both mitotic blockers and drivers, induced substantial cell death, mainly through apoptosis, in 2D and 3D cultures. Crucially, these synergistic concentrations were less toxic to non-tumor cells. This highlights the significance of combining BH3-mimetics with antimitotics, either blockers or drivers, which have reached the clinical trial phase, to enhance their effectiveness.

## 1. Introduction

Microtubule-targeting agents (MTAs) such as paclitaxel are widely adopted first-line chemotherapeutic agents in cancer clinical treatments, including for lung cancer. In fact, non-small cell lung cancer (NSCLC), the most prevalent form of lung cancer, is the leading cause of cancer death worldwide, resulting in one of the greatest public health challenges [1,2,3]. Approximately 350 people die each day from lung cancer and, in 2020, more than 2 million new lung cancer cases were reported [4,5]. By disrupting proper microtubule dynamics, MTAs lead to abnormal mitotic spindle assembly and compromise proper attachments of chromosomes to spindle microtubules, resulting in a chronic activation of the spindle assembly checkpoint (SAC), which eventually leads to cell death [6]. However, MTAs are associated with toxicity, as well as intrinsic and acquired resistance [7,8]. A plethora of second-generation antimitotic agents have thus been developed, including drugs targeting mitotic kinesins such as kinesin-like protein 1 (KLP1) and centromere protein E (CENPE), and mitotic kinases such as monopolar spindle 1 (MPS1), Aurora kinase B, and polo-like kinase 1 (PLK1), primarily involved in SAC signaling [9]. Unlike microtubules, which function in both mitosis and interphase, the role of these targets is primarily confined to mitosis. This limitation in function is expected to lead to lower toxicity compared to MTAs. Unfortunately, despite several of these antimitotic drugs advancing to clinical trials, they have shown unsatisfactory outcomes in cancer treatment, as monotherapy [9,10,11].

The effectiveness of antimitotic agents is hampered by the unpredictable outcomes of cancer cells during extended mitotic arrests [12,13]. Cancer cells arrested in mitosis may either die within this phase, or undergo slippage, wherein they exit mitosis without undergoing division. Slippage is primarily driven by the gradual degradation of cyclin B, even in the presence of an active SAC, which ultimately leads to mitotic exit. The determining factor between cell death in mitosis or slippage is the relative rates of cyclin B degradation and apoptotic signal accumulation [14,15,16]. According to these two competitive network models, in cases where cyclin B levels decline below the mitotic exit threshold before the accumulation of death signals reaches the necessary level for initiating apoptosis, slippage takes place. Conversely, if the death signals exceed the threshold required to trigger cell death before cyclin B levels diminish enough to prompt mitotic exit, the cells undergo cell death during mitosis [13,17]. Cells that have slipped into mitosis can proceed along one of three distinct routes: they might undergo post-slippage death, enter a state of senescence, or continue dividing, consequently contributing to tumor growth [12,13,18]. Therefore, slippage is recognized as a major resistance mechanism against antimitotic agents [14].

Cell death induced by antimitotic agents occurs through the activation of the intrinsic mitochondrial apoptotic pathway. This pathway is controlled by the anti-apoptotic proteins (BCL-2, BCL-W, BCL-XL, and MCL-1), pro-apoptotic proteins (BAX and BAK), and BH3-only proteins (e.g., BAD, BIK, BIM, BID, and NOXA) [19]. BH3-mimetics are a new class of pro-apoptotic anti-cancer drugs that target the intrinsic mitochondria-dependent apoptotic signaling pathway, showing promising clinical results, especially in patients with hematologic malignancies [20,21].

Hence, it becomes possible to exert deliberate control over the interplay between the two competitive pathways, namely, cyclin B degradation and apoptosis signal accumulation. For instance, combining BH3-mimetics with antimitotics should shift the balance from slippage to cell death in mitosis or post-mitosis, thereby enhancing the effectiveness of the antimitotic agents. This approach has recently undergone testing, yielding promising preclinical results [22,23,24].

In this study, we evaluated the effectiveness of this strategy by combining the BH3-mimetic navitoclax, known for its high affinity towards BCL-2 anti-apoptotic proteins such as BCL-2, BCL-W, and BCL-XL, along with two second-generation antimitotic agents: BAY1217389, a selective MPS1 inhibitor representing mitotic drivers, and GSK923295, a selective CENPE inhibitor representing mitotic blockers [25,26]. The approach was assessed using lung cancer cells cultured in both traditional 2D settings and a three-dimensional (3D) cancer model, serving as a preclinical system to mimic physiological drug responses. Additionally, we undertook a mechanistic study to understand how navitoclax promotes cancer cell death when combined with the antimitotic agents BAY1217389 or GSK923295. We found that the combination of the BH3-mimetic with both GSK923295 and BAY1217389 induced significant cell death, primarily through apoptosis, in both 2D and 3D cultures. Importantly, synergistic concentrations exhibited lower toxicity towards non-tumor cells. This underscores the relevance of combining BH3-mimetics with antimitotics, specifically CENPE and MPS-1 inhibitors, which have advanced to the clinical trial phase, to amplify their efficacy.

## 2. Materials and Methods

### 2.1. Small Molecule Inhibitors

Inhibitors of CENPE (GSK923295), BCL-2/BCL-XL (navitoclax), and MPS-1 (BAY1217389) were obtained from MedChem Express (Shanghai, China) and were reconstituted in sterile dimethyl sulfoxide (DMSO, Sigma-Aldrich Co., Ltd., St. Louis, MO, USA) to a stock concentration of 5 or 10 mM. Several aliquots were prepared and stored at −20 °C to avoid repeated cycles of freezing and thawing and, consequently, the loss of compounds’ activity. For each independent experiment, a work solution was prepared in fresh culture medium to prepare the desired concentrations.

### 2.2. Cell Culture

A549 (Human Lung Adenocarcinoma; American Type Culture Collection) and NCI-H460 (Human large cell lung cancer; European Collection of Cell Culture) cancer cell lines were grown in RPMI-1640 (Roswell Park Memorial Institute, Biochrom, Buffalo, NY, USA) and DMEM (Dulbecco’s Modified Eagle’s, Biochrom) culture medium, respectively, and supplemented with 10% of heat-inactivated fetal bovine serum (FBS, Biochrom, Berlin, Germany) and 1% of Pen/Strep (Biochrom). In addition, DMEM was supplemented with 1% of non-essential amino acids (Sigma-Aldrich Co., Ltd.). The non-cancer cell line HPAEpiC (Human Pulmonary Alveolar Epithelial Cells; ScienCell Research Laboratories, San Diego, CA, USA) was grown in the same conditions as A549. All cell lines were maintained in a cell incubator at 37 °C with 5% CO_2_ (Hera Cell, Heraeus, Hanau, Germany) with a humidified atmosphere.

### 2.3. RNA Extraction, cDNA Synthesis and Quantitative Real-Time PCR

Total RNA extraction and cDNA synthesis were performed as previously reported [18]. DNA was amplified using iQ™ SYBR Green Supermix Kit (Bio-Rad, Laboratories, Inc., Hercules, CA, USA) on an iQ Thermal Cycler (Bio-Rad), according to the following program: initial denaturing step at 95.0 °C for 3 min; 40 cycles at 94.0 °C for 20 s; 62.0 °C for 30 s and 72.0 °C for 30 s. The melt curve included temperatures from 65.0 to 95.0 °C, with increments of 0.5 °C for 5 s. The primers, at 10 µM, were as follows: MPS-1: forward 50-TCAAGGAACCTCTGGTGTCA-30 and reverse 50-GGTTACTCTCTGGAACCTCTGGT-30; CENPE: forward 50-GTCGGACCAGTTCAGCCTGATA-30 and reverse 50-CCAAGTGATTCTTCTCTGCTGTTC-30; GAPDH: forward 50-ACAGTCAGCCGCATCTTC-30 and reverse 50-GCCCAATACGACCAAATCC-30; Actin: forward 50-AATCTGGCACCACACCTTCTA-30 and reverse 50-ATAGCACAGCCTGGATAGCAA-30. The data were acquired using CFX ManagerTM Software (version 1.0, BioRad) and the results were analyzed according to CT and normalized against Actin and GAPDH expression levels, which were used as housekeeping genes.

### 2.4. Protein Extracts and Western Blotting

Protein extracts’ pelleted cells were resuspended in lysis buffer (50 mM Tris pH 7.5; 150 mM NaCl; 1 mM EDTA; 1% Triton-100) containing a protease inhibitor cocktail (Sigma-Aldrich). A BCA^TM^ Protein Assay Kit (Pierce Biotechnology, Rockford, IL, USA) was used for protein quantification according to the manufacturer’s instructions. For MPS1 detection, a total of 20 µg of protein lysate was resuspended in SDS-sample buffer (375 mM Tris pH 6.8; 12% SDS; 60% Glycerol; 0.12% Bromophenol Blue; 600 nM DTT) and boiled for 3 min, and proteins were separated on a 7.5% SDS–PAGE gel. Following SDS-PAGE, proteins were transferred onto nitrocellulose membranes (Amersham) using the Trans-Blot Turbo Transfer System (Bio-Rad). For Cyclin B1 detection, the same procedure was followed, but using 10 μg of total protein lysate and a 10% SDS–PAGE gel. For CENPE detection, a total of 60 µg of protein lysate was also resuspended in SDS-sample buffer and boiled for 3 min, and was resolved on a 4–20% gradient gel (Bio-Rad). Then, proteins were transferred onto nitrocellulose membranes (Amersham) using wet-tank transfer via a Mini Trans-Blot Electrophoretic Transfer Cell (Bio-Rad). Then, membranes were blocked in 5% of non-fat dried milk (*w*/*v*) dissolved in TBST (50 mM Tris pH 7.5; 150 mM NaCl, 0.05% Tween-20), and were incubated overnight at 4 °C with the following primary antibodies diluted in TBST: mouse anti-α-tubulin (1:5000, T568 Clone B-5-1-2, Sigma-Aldrich), rabbit anti-cyclin B1 (1:500, C8831, Sigma-Aldrich), mouse anti-CENPE (1:250, (C-5): sc-376685, Santa Cruz Biotechnology, Heidelberg, Germany), and mouse anti-MPS-1 (1000, (N1): sc-56968, Santa Cruz Biotechnology, Heidelberg, Germany). The membranes were washed three times in TBST, and then incubated for 1 h with appropriate horseradish-peroxidase-conjugated secondary antibodies (1:1500 (anti-mouse) or 1:1000 (anti-rabbit), Vector). Proteins were detected using the Enhanced Chemiluminescence (ECL) method in a ChemiDOc (Bio-Rad). The protein signal intensity quantification was performed using Image Lab 6.1v software and normalized against α-tubulin expression levels.

### 2.5. Indirect Immunofluorescence

A total of 0.09 × 10^6^ cells/mL were grown on poly-L-lysine-coated coverslips in complete culture medium for 24 h. Subsequently, cells underwent treatment with MPS-1 and CENPE inhibitors. After 24 h, they were fixed in methanol (Sigma-Aldrich, Co., Ltd., Gillingham, UK) at −20 °C for 10 min, followed by three washes with PBS for 5 min each. The cells were then blocked using 10% FBS in PBST (0.05% Tween-20 in PBS) for 30 min at room temperature. Then, cells were subjected to a 1 h incubation with primary antibodies (mouse anti-α-tubulin, 1:2500, Sigma-Aldrich Co., Ltd., Gillingham, UK; human anti-CREST, 1:3000, gifted by E. Bronze-da-Rocha, IBMC, Porto, Portugal) diluted in 5% FBS in PBST. After three washes in PBST, cells were exposed to Alexa Fluor 488-conjugated secondary antibody (1:1500, Molecular Probes, Eugene, OR, USA). To visualize the DNA, cells were stained with 2 µg/mL 4′,6-diamidino-2-phenylindole (DAPI, Sigma-Aldrich) diluted in Vectashield mounting medium (Vector, H-1000, Burlingame, CA, USA).

### 2.6. MTT Viability Assay

To measure cell viability, a tetrazolium salt 3-(4, 5-dimethylthiazol-2-yl)-2,5-diphenyltetrazolium bromide (MTT) assay was used. A total of 0.05 × 10^6^ cells/mL of A549 or NCI-H460 cells and 0.065 × 10^6^ cells/mL of HPAEpiC cells were seeded into a 96-well plate in complete media, allowing them to adhere overnight prior to drug exposure. After 24 h, the culture medium was replaced with fresh medium containing 2-fold serial dilutions of the inhibitors ranging from 62.5 nM to 1000 nM for GSK923295, 500 nM to 8000 nM for BAY1217389, and 1000 nM to 16,000 nM for Navitoclax. Forty-eight hours later, 20 µL of tetrazolium salt MTT (5 mg/mL PBS) was added to 200 µL of fresh medium for 4 h. The formazan crystals were dissolved in 100 µL of DMSO, and the optical density was retrieved at 570 nm using a microplate reader (Biotek Synergy 2, Winooski, VT, USA) coupled to Gen5 software (version 1.07.5, Biotek, Winooski, VT, USA). GraphPad Prism version 8 (GraphPad software Inc., San Diego, CA, USA) was used to calculate the mean 50% inhibition concentration (IC_50_) values. Additionally, the combined treatment effects were evaluated using a dual-drug crosswise concentration via Combenefit Software (version 2.021, Cancer Research UK Cambridge Institute, Cambridge, UK).

### 2.7. Apoptosis Detection

#### 2.7.1. TUNEL Assay

To detect apoptosis, the DeadEnd Fluorometric TUNEL System (Promega, Madison, WI, USA) was used following the manufacturer’s guidelines. For DNA staining, 2 mg/mL of DAPI in Vectashield mounting medium was used. The extent of cell death was evaluated by counting TUNEL-positive cells among a total of 500 cells, from at least 10 random microscopic fields, for each experimental condition under a fluorescence microscope.

#### 2.7.2. Annexin V/PI Staining

Apoptotic cell death was assessed using the Annexin V-FITC Apoptosis Detection Kit (eBioscience, Vienna, Austria) according to the manufacturer’s instructions. Briefly, 0.09 × 10^6^ cells/mL were seeded into 6-well plates, and 24 h later cells were treated with MPS1/CENPE inhibitors alone or in combination with navitoclax at the concentration of the respective synergistic points. After 48 h, both floating and adherent cells were gathered and pelleted by centrifugation at 1000 rpm for 5 min, then suspended in binding buffer 1×. Subsequently, Annexin V-FITC was added and allowed to incubate for 10 min, shielded from light. After washing, cells were once again resuspended in binding buffer 1×, and 20 μg/mL of Propidium iodide (PI) was added. Fluorescence analysis was performed using the BD Accuri™ C6 Plus Flow cytometer (BD Biosciences, Qume Drive, San Jose, CA, USA), and the data were processed using BD Accuri TM C6 Plus software, version 1.0.27.1. At least 20,000 events per sample were collected.

To evaluate apoptotic cell death in 3D cultures, after 48 h of treatment with MPS1/CENPE inhibitors alone or in combination with navitoclax at the concentration of the respective synergistic points, approximately 32 spheroids were collected from a 96-well ultra-low attachment plate and transferred to a 15 mL centrifuge tube. Once the spheroids precipitated, the supernatant was removed, and PBS was added to wash the spheroids. After, PBS was removed and 200 μL of trypsin (GIBCO, Invitrogen, Waltham, MA, USA) was added. The spheroids were then incubated at 37 °C for 25 min to guarantee their total dissociation into single cells. After addition of 500 μL of culture medium, the cells were centrifuged at 1000 rpm for 4 min and washed with PBS. The samples were treated with an “Annexin V-FITC Apoptosis Detection Kit” according to the manufacturer’s instructions. At least 20,000 events per sample were collected.

### 2.8. Mitotic Index Determination

A total of 0.09 × 10^6^ cells/mL were grown in six-well dishes and treated for 24 h with CENPE inhibitor alone or in combination with navitoclax at the concentration of the respective synergistic point. Cells treated with 1 µM of microtubule depolymerizing agent Nocodazole were used as the positive control for antimitotic activity. Untreated cells and cells treated with DMSO, to assess compound solvent-mediated cytotoxicity, were also included as controls. The mitotic index, the percentage of mitotic cells over a total cell population, was determined by cell rounding under phase-contrast microscopy. At least 3000 cells were counted from random microscope fields.

### 2.9. Time-Lapse Microscopy

For live-cell imaging, a total of 0.09 × 10^6^ A549 cells were seeded into a LabTek II chambered cover glass (Nunc, Penfield, NY, USA) in complete RPMI culture medium. Sterile water was added to the remaining wells to guarantee a humidified atmosphere. The cells were incubated overnight at 37 °C under 5% CO_2_. Then, the medium was replaced by fresh medium in the presence of MPS-1/CENPE inhibitors alone or in combination with navitoclax at the concentration of the respective synergistic points. Time-lapse images were taken every 5 min over a 48 h period using differential interference contrast (DIC) optics, and a 63× objective on an Axio Observer Z.1 SD inverted microscope (Carl Zeiss, Oberkochen, Germany). The microscope is equipped with an incubation chamber set to 37 °C and 5% CO_2_. ImageJ software (version 1.47, Rasband, W.S., ImageJ, U. S. National Institutes of Health, Bethesda, MD, USA) was used to create movies from the time-lapse images.

### 2.10. Phase-Contrast and Fluorescence Microscopy Images

Phase-contrast microscopy images were obtained using a Nikon TE 2000-U microscope (Nikon, Amsterdam, The Netherlands) equipped with a 10× objective and connected to a DXM1200F digital camera running Nikon ACT-1 software version 2.63 (Melville, NY, USA). Fluorescence imaging was acquired using an Axio Observer Z.1 SD microscope, coupled with an AxioCam MR3 and the Plan Apochromatic 100×/NA 1.4 objective, and the images were processed using ImageJ.

### 2.11. Colony Formation Assay

A total of 500 A549 cells were seeded in six-well plates, allowed to attach for 24 h, and treated with drugs in monotherapy or in combination. Untreated and DMSO-treated cells were also included. Forty-eight hours later, cells were washed twice with PBS and incubated in a drug-free DMEM medium for 7 days. After this period, the colonies were fixed for 25 min using 100% methanol at −20 °C and then stained for 20 min with 0.05% (*w*/*v*) violet crystal (Merck, Rahway, NJ, USA) in distilled water. The count of colonies for each condition was derived from three independent experiments. Plating efficiency (PE) was calculated as the percentage of the number of colonies that grew compared to the number of cells seeded in the control. Additionally, the survival fraction for each condition was calculated as the number of colonies over the number of cells seeded × 1/PE.

### 2.12. Caspase Activity Assay

To evaluate caspase-9 activity, cells were seeded as described for immunofluorescence assay. Following a 24 h incubation with mitotic inhibitor, navitoclax, alone or in combination, the medium was aspirated, and cells were washed with PBS. Subsequently, 150 µL of Glo Lysis Buffer (Promega, Madison, WI, USA) was added to each well, and the cells were incubated for 5 min at room temperature. Caspase-9 activity was determined as previously described [27]. For caspase-9 detection, lysates (10 mL) were mixed with 200 mL of assay buffer (100 nM HEPES (pH 7.5), 20% (*v*/*v*) glycerol, 5 mM DTT, 0.5 mM EDTA) followed by incubation at 30 °C for 30 min. After incubation, the reaction was started by adding 10 mL of caspase-9 fluorogenic substrate N-acetyl-Leu-Glu-His-Asp 7-amido-4-trifluoromethylcoumarin (Sigma-Aldrich) at a final concentration of 180 mM. Fluorescence was determined using a microplate reader (Biotek Synergy 2) to 400 nm excitation and 500 nm emission, in a kinetic reaction for 5 min. The obtained results were normalized against the protein content. For each assay, normalization was also carried out against the value obtained in the untreated group, with a reference value set at 1.

### 2.13. Spheroid Formation, Drug Treatment and Viability Assay

The generation of A549 spheroids, drug treatment, and the assessment of spheroids viability were performed as previously reported [22]. Briefly, 4000 cells/well were seeded into 96-well ultra-low attachment plates and 4 days later treated with MPS1/CENPE inhibitors alone or in combination with navitoclax at the concentrations of 4000–16,000 nM. After 48 h, the spheroid viability was determined via CellTiter-Glo 3D cell viability assay (Promega) according to the manufacturer’s instructions.

### 2.14. Statistical Analysis

All assays were performed in triplicate from at least three independent experiments. Data are expressed as mean ± standard deviation (SD), and statistical analysis was carried out in GraphPad Prism Software Inc. v8 using the unpaired *t*-test or two-way ANOVA with Tukey’s multiple comparison test; values of * *p* < 0.05, ** *p* < 0.01, *** *p* < 0.001, and **** *p* < 0.0001 were considered statistically significant.

## 3. Results

### 3.1. CENPE and MPS1 Are Overexpressed in Lung Cancer Cells

CENPE is a microtubule-dependent plus-end-directed motor belonging to the kinesin-7 subfamily and is crucial for the congression of initially misaligned chromosomes [27]. Inhibitors of CENPE lead to chromosome misalignment, resulting in an extended mitotic arrest, acting as mitotic blockers, and ultimately leading to cell death in mitosis [28]. MPS-1 is a protein kinase and a crucial activator of the SAC [29]. Inhibitors of MPS1 override the SAC and induce premature mitotic exit, leading to massive chromosome missegregation and eventual cell death, acting as mitotic drivers [30,31]. Both MPS1 and CENPE have been reported to be overexpressed in cancer cells, making them potential targets for cancer therapy [32,33,34,35].

Thus, we first examined the expression of CENPE and MPS-1 in A549 and NCI-H460 non-small cell lung cancer (NSCLC) cell lines. The results demonstrated that CENPE and MPS1 mRNA levels, determined by qRT-PCR, were upregulated in both lung cancer cell lines when compared to the non-tumor cell line HPAEpiC (Figure 1a,c). Western blot analysis also evidenced an increase in protein levels for both targets (Figure 1b,d). The findings align with prior studies that have documented the increased expression of CENPE and MPS1 in lung cancer, underscoring the significance of targeting these proteins [33,36].

Due to the fact that A549 cells represent a model of NSCLC, the most common lung cancer type, and exhibit elevated protein expression of CENPE and MPS1 compared to the large cell lung cancer model NCI-H460 cells, we selected A549 cells for the subsequent experiments in this study [37].

### 3.2. Navitoclax Synergizes with the Mitotic Blocker GSK923295 and the Mitotic Driver BAY1217389 in Killing Lung Cancer Cells

To assess whether the BH3-mimetic navitoclax potentiates the antiproliferative activity of the CENPE inhibitor GSK923295 or the MPS1 inhibitor BAY1217389, we initially examined cellular cytotoxicity using the MTT assay, after exposure of A549 cells to these compounds, individually and in combination for 48 h. A dual-drug concentration crosswise matrix was performed for each combination, covering a concentration range from 0 to 16,000 nM for navitoclax, 0 to 1000 nM for GSK923295, and 0 to 8000 nM for BAY1217389. Using the Combenefit Software, we assessed the percentage of viable cells (Figure 2a,b) and calculated the combinatorial interaction effect score (Figure 2c,d). This analysis allowed us to determine the IC_50_ of each compound (Table 1 and Table 2). 

The IC_50_ values of GSK923295 and BAY1217389 were 150.0 ± 30 nM and 4340.0 ± 60 nM, respectively, while that of navitoclax exceeded 13,000 nM, indicating its lower cytotoxicity. Interestingly, a synergistic effect was observed with both GSK923295 + navitoclax and BAY1217389 + navitoclax combinations (Figure 2c,d). The synergistic combination with the lowest concentrations (1000 nM of navitoclax with 125 nM of GSK923295, and 1000 nM of navitoclax with 500 nM of BAY1217389) were selected for subsequent experiments. It is noteworthy that the combination of 1000 nM of navitoclax with 125 nM of GSK923295 corresponds to 13- and 1.2-fold less than their respective IC_50_ values, while the combination of 1000 nM of navitoclax with 500 nM of BAY1217389 is approximately 13- and 8-fold less than their respective IC_50_ values. This is particularly relevant for minimizing toxicity and side effects reported in clinical trials for these drugs [9,38]. Interestingly, the selected concentrations of both GSK923295 + navitoclax and BAY1217389 + navitoclax did not significantly affect the viability of the non-cancer HPAEpiC cells (Appendix A). This suggests that cancer cells are more responsive to these treatments than non-cancer cells.

We also performed a colony formation assay in A549 cancer cells treated with the GSK923295 + navitoclax or BAY1217389 + navitoclax combinations. For this, after 48 h of drug exposure, the medium was replaced with fresh medium and A549 cells were maintained for 7 days in a cell culture incubator at 37 °C with 5% CO_2_. On the 7th day, the colonies were counted. Our results showed that both combinations of GSK923295 + navitoclax and BAY1217389 + navitoclax were able to reduce colony formation when compared to single treatments (Figure 2e–g). Indeed, a reduction to 8.3 ± 4.2% of colony survival fraction after GSK923295 + navitoclax combination exposure was observed when compared to GSK923295 (30.0 ± 5.9%) and navitoclax (88.0 ± 7.1%) monotherapy, and a reduction to 4.4 ± 3.0% after BAY1217389 + navitoclax treatment compared to BAY1217389 (12.2 ± 3.6%) and navitoclax (86.7 ± 5.7%) drugs alone. These results suggest that the combinatorial approaches exhibit an ability to maintain long-term cellular cytotoxicity, preventing the proliferation of cancer cells.

Therefore, the BH3-mimetic synergizes with both the antimitotic agent that induces a SAC-mediated mitotic block and the antimitotic agent that drives cells through an aberrant mitosis by overriding the SAC. We next proceeded further with the selected combinations to obtain a deeper understanding of the cellular mechanism behind their synergistic cytotoxicity.

### 3.3. Navitoclax Prevents Mitotic Slippage Caused by GSK923295 Treatment by Accelerating Cell Death during Mitosis

To gain insights into the cellular mechanisms underlying the synergistic cytotoxicity of the GSK923295 + navitoclax combination, A549 cells were exposed to these combinations, single agents, or medium/DMSO (controls) for 48 h, and subsequently examined using phase-contrast microscopy. Upon treatment with GSK923295 alone, as expected, we observed an accumulation of mitotic cells, similar to the effect induced by Nocodazole, a well-known antimitotic agent used here as a positive control (Figure 3a,b). This observation was further confirmed by calculating the mitotic index, which was significantly higher in GSK923295-treated cell cultures (66.4 ± 6.6%) compared to untreated (8.7 + 0.5%) and DMSO-treated cells (6.6 + 1.8%) (Figure 3b). Treatment with navitoclax alone did not significantly impact normal cell cycling, except for a few instances of cell death. Interestingly, when GSK923295 was combined with navitoclax, the mitotic index decreased to 47.0 ± 2.4%, but it still remained higher compared to the controls. The complex formed by cyclin B1 and cyclin-dependent kinase 1 (CDK1) acts in the regulation of mitotic entry and progression. Cyclin B1 degradation is essential for mitotic exit [39]. Thus, high levels of cyclin B are indicative of a mitotic arrest. The results show an increase in cyclin B1 protein levels after GSK923295 exposure (Figure 3c), as expected, and are complementary to the mitotic index determination. Conversely, both the combinations of GSK923295 + navitoclax and BAY + navitoclax, as well as BAY and navitoclax alone, had no significant effect on cyclin B1 protein levels.

We then carried out time-lapse microscopy to track live cells treated with the GSK923295 + navitoclax combination, allowing us to monitor their spatiotemporal dynamics. Our objective was to unveil the fate of the cells arrested in mitosis due to GSK923295 or GSK923295 + navitoclax treatment. A549 tumor cells were treated with GSK923295 and navitoclax, alone or in combination, and each cell was followed over 48 h via live cell time-lapse microscopy.

Control cells completed mitosis within 28.2 ± 5.9 min, and the presence of navitoclax did not significantly affect this timing (26.8 ± 5.1 min) (Figure 3d). However, treatment with GSK923295 led to a substantial extension of mitosis duration, lasting an average of 952.1 ± 4165.0 min. Interestingly, when navitoclax was added to GSK923295, it significantly reduced the duration of mitotic arrest by more than 2.5-fold, suggesting that navitoclax accelerates cell death (Figure 3d). Regarding the cell fates, we observed that most of the navitoclax-treated cells underwent normal cell division, except for a few cells (2.9 + 4.0%) that experienced post-mitotic death (PMD). This suggests that navitoclax alone does not exhibit toxicity to tumor cells, at least at the given concentration (Figure 3e,f, and Appendix A). For cells treated with GSK923295 alone, the majority (92.8%) of cells that arrested in mitosis managed to survive throughout the experiment, with only a small fraction (7.2 ± 8.8%) undergoing cell death in mitosis (DiM). Among the surviving cells, 54.0 ± 17.6% continued to survive after cell division (post-mitotic survival, PMS), and 38.8 ± 18.5% persisted after mitotic slippage (post-slippage survival, PSS) (Figure 3e,d, and Appendix A). In contrast, in cells subjected to the GSK923295 + navitoclax combination treatment, a substantial portion (95%) of the cells arrested in mitosis died, with the majority (88.1 ± 17.2%) of these deaths occurring during mitosis (DiM), and the remainder (6.9 ± 13.6%) occurring after cell division (Figure 3e,d, and Appendix A). Only a small fraction (5.0 ± 5.9%) managed to survive after completing mitosis (PMS). Notably, no cells underwent mitotic slippage. Therefore, when combined with GSK923295, navitoclax shifts cell fate from mitotic slippage to cell death in mitosis, thereby eliminating the possibility of cellular survival by slippage.

Apoptosis was the predominant mechanism of cell death in the GSK923295 + navitoclax combination, as evidenced by both flow cytometry analysis of Annexin V/PI-stained cells and immunostaining using the TUNEL assay (Figure 4a–d). In this context, to evaluate the possible apoptotic pathway associated with drug treatments, the activity of caspase-9 was assessed. Caspase-9, a critical initiator caspase essential for the intrinsic pathway of apoptosis, upon activation, cleaves and activates downstream effector caspases-3 and -7. These effectors, in turn, target key regulatory and structural proteins for proteolysis, leading to cell death [40,41]. The results showed an increase in caspase-9 activity after GSK923295 + navitoclax combination treatment (4.6 ± 0.2), compared to GSK923295 (1.9 ± 0.2) and navitoclax (1.1 ± 0.3) drugs alone (Figure 4e), indicating that compounds target the intrinsic apoptosis pathway.

### 3.4. Navitoclax Prevents Post-Mitotic Survival Induced by BAY1217389 by Enhancing Post-Mitotic Death, but Only Partially

We also conducted time-lapse microscopy to monitor live cells treated with the combination of BAY1217389 and navitoclax, aiming to gain mechanistic insights into the cytotoxicity of this combination. Two-dimensional cultures of A549 cancer cells were subjected to treatment with BAY1217389 and navitoclax, either individually or in combination. Each cell was observed over 48 h using live cell time-lapse microscopy. The duration of mitosis in untreated cells was 28.2 ± 5.9 min, and navitoclax treatment did not significantly alter this duration (26.8 ± 5.1 min) (Figure 5a). In contrast, BAY1217389 treatment notably reduced mitosis duration to 17.7 ± 4.3 min, accelerating mitotic exit, as expected for a mitotic driver. Cotreatment with BAY1217389 and navitoclax did not significantly impact this phenotype, resulting in a mitosis duration of 15.5 ± 4.2 min. In terms of cell fate, we observed that the majority of navitoclax-treated cells underwent normal cell division, with only a few cells (2.9 ± 4.0%) experiencing PMD, again suggesting that, at least during the 48 h time course, navitoclax alone does not exhibit toxicity to cancer cells (Figure 5b,c). Treatment with BAY1217389 primarily induced PMS (75.1 ± 20.5%) and only a small percentage of PMD (6.0 ± 7.9%). Interestingly, cotreatment with BAY1217389 and navitoclax led to a significant increase in PMD (49.2 ± 11.3%, *p* < 0.0001), along with an almost 2-fold and significant decrease in PMS (39.1 ± 5.3%, *p* < 0.0001). However, the percentage of PMS cells is still significant and raises concerns about the efficient eradication of cancer cells. Therefore, navitoclax only partially sensitizes cancer cells to BAY1217389, as a significant fraction of cancer cells still escape cell death.

Cell death in the BAY1217389 + navitoclax combination was primarily attributed to apoptosis, as demonstrated by both flow cytometry analysis of Annexin V/PI-stained cells and immunostaining using the TUNEL assay (Figure 6a–d). The caspase-9 activity also was increased after BAY1217389 + navitoclax treatment (3.2 ± 0.4) when compared to BAY1217389 (1.9 ± 0.5) and navitoclax (1.2 ± 0.1) drugs alone (Figure 6e), indicating an intrinsic apoptotic pathway participation.

### 3.5. Navitoclax Sensitizes 3D Lung Cancer Spheroids to GSK923295 and BAY1217389 Treatment

Considering the substantial synergistic effect observed in a 2D system, we evaluated the effectiveness of the GSK923295 + navitoclax and BAY1217389 + navitoclax combinations in a 3D spheroid model. This model mimics tumor architecture and the microenvironment, making it a relevant in vitro preclinical model for cancer [42].

A dual-drug concentration crosswise matrix was made encompassing various different compound concentrations ranging from 0 to 16,000 nM. After 48 h of mono- or combination treatments, the spheroids’ viability was determined by CellTiter-Glo assay and the IC_50_ values for the tested compounds were determined (Table 3 and Table 4). 

The IC_50_ for both GSK923295 and BAY1217389 was >16,000 nM. In contrast, the IC_50_ for navitoclax was approximately 6400 nM in the spheroids, suggesting that lung cancer cells exhibit lower sensitivity to GSK923295 and BAY1217389 but higher sensitivity to navitoclax in a 3D cell culture system compared to a 2D system (Figure 7a,c,d,f). The combination of GSK923295 and BAY1217384 with navitoclax led to significantly greater cell death in A549 lung cancer cells compared to individual treatments, emphasizing the synergistic effect of the combinations (Figure 7b,e).

Macroscopic examination of the spheroids treated with the lowest synergistic concentration of the GSK923295 + navitoclax and BAY1217389 + navitoclax combinations (4000 nM of navitoclax with 4000 nM of GSK923295 or BAY1217389) revealed a loosely compacted and partially fragmented structure (Figure 8a,d). Many cells had lost adhesion to the spheroid surface, indicating cytotoxic effects compared to the intact control spheroids. Additionally, cell death in both combinations was primarily attributed to apoptosis, as demonstrated by flow cytometry analysis of Annexin V/PI-stained cells (Figure 8b,c,e,f). Indeed, the combination of GSK923295 + navitoclax enhanced cell death by apoptosis (61.2 ± 11.7%) when compared to GSK923295 (40.7 ± 0.4%) and navitoclax (48.4 ± 1.2%) alone treatments (Figure 8b,c). Similarly, BAY1217389 + navitoclax significantly increased the Annexin-V positive cells (62.5 ± 5.1%) compared to BAY1217389 (24.8 ± 6.7%) or navitoclax (42.8 ± 7.1%) (Figure 8e,f).

Overall, similar to the effect on 2D cancer cultures, the combination of the antimitotics GSK923295 and BAY1217389 with the anti-apoptotic inhibitor navitoclax enhances cancer cell death in a model that mimics a solid in vivo tumor. In contrast to the 2D results, where BAY1217389 + navitoclax combinations had only partial cytotoxic activity, we found that both GSK923295 + navitoclax and BAY1217389 + navitoclax combinations exhibited similar cytotoxic activity on spheroids derived from A549 lung cancer cells in the 3D model, although GSK923295 alone was more efficient in spheroid cell killing than BAY1217389.

## 4. Discussion

This study aimed to investigate the potential of combining the BH3-mimetic navitoclax with the mitotic blocker CENPE inhibitor GSK923295 or the mitotic driver MPS1 inhibitor BAY1217389 in 2D and 3D in vitro models of NSCLC. Our results demonstrate a synergistic cytotoxic activity against 2D cancer cell culture with both combinations, with significant degrees of cell death induced, mainly by apoptosis, which was more pronounced after GSK923295 + navitoclax than the BAY1217389 + navitoclax combinatorial treatment. Interestingly, the discrepancy between the two combinations was dissipated in the context of the 3D spheroid model, suggesting that the inhibition of CENPE or MPS1 in combination with BH3-mimetics is worth further investigation in the clinic as a potent therapeutic option.

The combination yielded increased cancer cell killing in both 2D and 3D culture systems, although establishing the IC_50_ of GSK923295 and BAY1217384 in 3D cultures proved challenging. This limitation may stem from the intricacies inherent in working with 3D spheroids. However, it also provides insight into why these drugs did not achieve success in clinical trials when used as monotherapy.

In the realm of drug testing, the divergence between 2D and 3D cellular models has yielded intriguing insights. It is commonly observed that the IC_50_ values of chemotherapeutic drugs are higher in 3D spheroids compared to their 2D counterparts. This phenomenon is expected due to the structural complexity of spheroids, mimicking the real tumor. Nonuniform growth and oxygen gradients with hypoxic cores and diffusion gradients similar to those in vivo can hinder the effective penetration of drugs into the cells, necessitating higher drug concentrations to elicit cell death [43,44,45]. Supporting the literature, our findings revealed that 3D spheroids require higher concentrations of GSK923295 and BAY1217389 to effectively inhibit cancer cells compared to the 2D system. On the contrary, we observed greater sensitivity to navitoclax in the 3D cell culture system when compared to the 2D system. This less common phenomenon could be attributed to distinct features of the drug, the cells, or the experimental design. Notably, in the presence of navitoclax, 3D spheroids exhibited a heightened synergistic response to both GSK923295 and BAY1217389, reaffirming the synergy observed in the 2D cell culture system. This underscores the potential for the 3D culture system’s characteristics to facilitate synergistic interactions among the tested drugs.

Despite the observed differences between 2D and 3D systems, the study contributes to understanding the complexities of drug responses, and lays the groundwork for future investigations into the combination of navitoclax with GSK923295 and BAY1217389 in 3D and in vivo settings.

The BH3-mimetic navitoclax demonstrates synergistic effects with both GSK923295 and BAY1217389 in cancer cell killing. Our live-cell imaging analysis reveals distinct mechanisms for this synergy. Specifically, navitoclax seems to prevent mitotic slippage induced by GSK923295 by expediting cell death during mitosis. In contrast, it hinders post-mitotic survival prompted by BAY1217389 by intensifying post-mitotic cell death. In both scenarios, navitoclax tips the balance from cell survival towards cell death, effectively eliminating the opportunity for cellular escape, a phenomenon often observed when mitotic blockers and drivers are used as monotherapy [6,9,22].

These results align with our previously published work targeting polo-like kinase 1 (PLK1) [22]. The cytotoxic activity of PLK1 inhibitors, acting as mitotic blockers, was enhanced when combined with the BH3-mimetic navitoclax or ABT-737. This further reinforces the successful strategy of combining antimitotics with prosurvival inhibitors.

In conclusion, the results of the present study demonstrate the potential therapeutic advantages of co-inhibiting BCL-2 family proteins alongside CENPE or MPS1. These results warrant additional investigation, including long-term cellular assays and in vivo murine experiments, to assess the feasibility of translating these treatment approaches to clinical trials. Furthermore, the observed effects of these drug combinations, which include reduced viability and colony-forming capacity in cancer cells, as well as a significantly higher rate of cell death when compared to non-cancer cells, provide a robust basis for further clinical exploration.

## Figures and Tables

**Figure 1 pharmaceutics-16-00056-f001:**
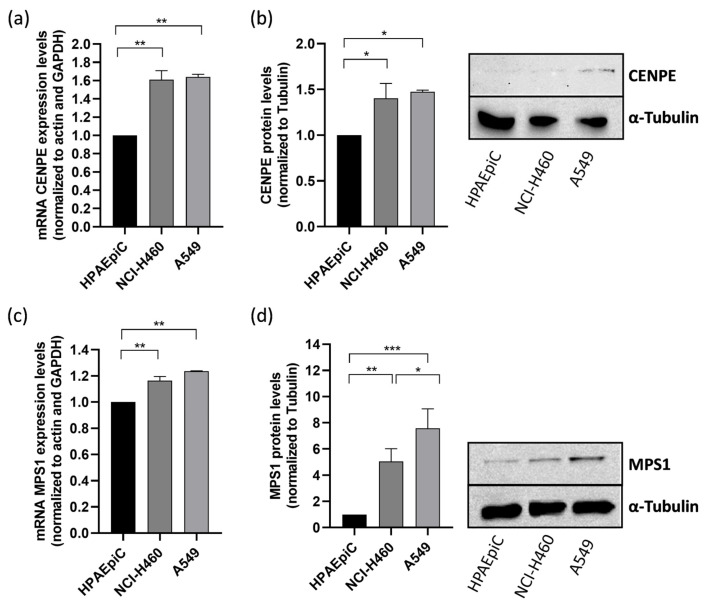
CENPE and MPS1 are overexpressed in NSCLC lung cancer cell lines. mRNA expression of CENPE (**a**) and MPS1 (**c**) was determined by qRT-PCR in A549 and NCI-H460 cancer cell lines, and was compared to that in non-tumor HPAEpiC cells. Protein levels of CENPE (**b**) and MPS1 (**d**) were quantified by Western blotting assay, using α-tubulin as control. Data represent the mean ± SD of three independent experiments, one-way ANOVA followed by Tukey’s multiple comparisons test. * *p* < 0.05; ** *p* < 0.01; *** *p* < 0.001.

**Figure 2 pharmaceutics-16-00056-f002:**
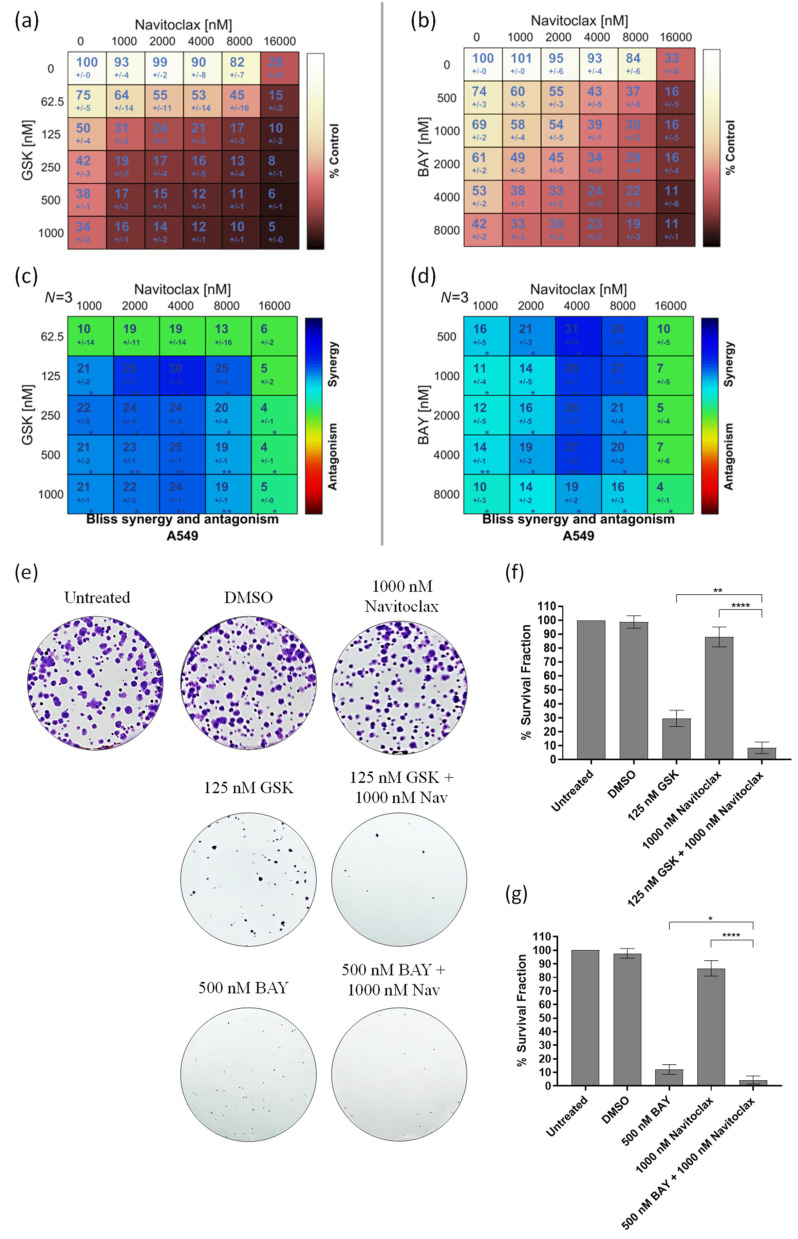
GSK923295 + navitoclax and BAY1217389 + navitoclax combinations potentiate cytotoxicity in 2D A549 lung cancer cell cultures. Cell viability (%) of single or combination therapies after 48 h of drug exposure (**a**,**b**), from three independent experiments as determined by MTT assay. Synergy scores calculated by the Bliss model of Combenefit software 2.021 with statistical relevance of * *p* < 0.05, ** *p* < 0.01, and *** *p* < 0.001. Asterisk indicates synergism effects (**c**,**d**). Colony formation assays were performed using A549 cells following 7 days (**e**). Quantification of survival fraction (%) after single or combination treatments as indicated (**f**,**g**). Data represent the mean ± SD of three independent experiments, one-way ANOVA followed by Tukey’s multiple comparisons test. * *p* < 0.05; ** *p* < 0.01; **** *p* < 0.0001.

**Figure 3 pharmaceutics-16-00056-f003:**
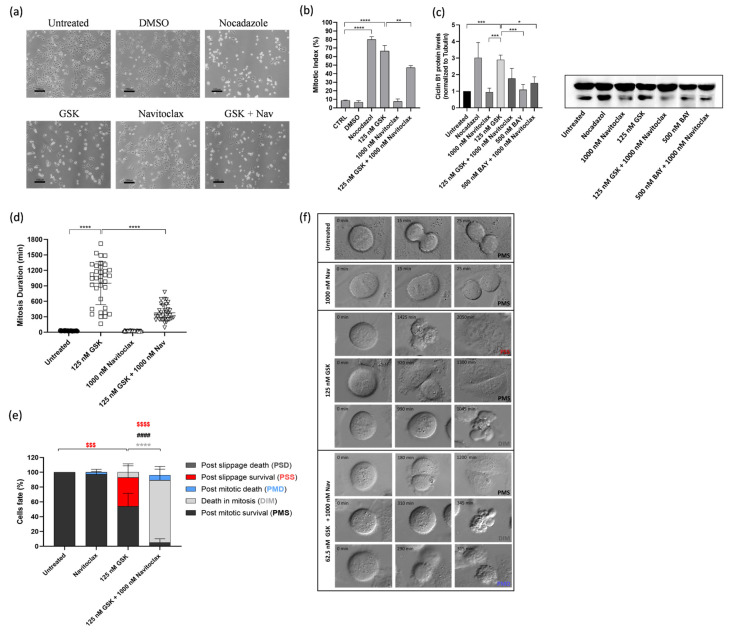
Combination of GSK923295 + navitoclax reduces mitotic arrest duration and prevents slippage by accelerating cell death in mitosis in lung cancer cells. Representative phase-contrate microscopy images after 24 h of drugs alone or in combination (**a**). Quantification of mitotic index; 0.25% DMSO (compound solvent) and 1 μM Nocodazole (mitotic blocker agent) were used as negative and positive controls, respectively (**b**). Quantification of mitosis duration after respective drug treatments via time-lapse microscopy (**c**). Cyclin B1 levels as determined by Western blotting assay. (**d**) Quantification of cell fate (%) for 48 h using different treatments as indicated (**e**). Representative time-lapse image sequences of A549 cells immediately after drug treatments (**f**). Data represent the mean ± SD of at least three independent experiments, one-way ANOVA followed by Tukey’s multiple comparisons test. * *p* < 0.05, ** *p* < 0.01; *** *p* < 0.001; **** *p* < 0.0001. $$$ difference (*p* < 0.001) of post-slippage survival cells (%) between untreated and 125 nM GSK923295. $$$$ difference (*p* < 0.0001) of post-slippage survival cells (%) between 125 nM GSK923295 and 125 nM GSK923295 + 1000 nM navitoclax. * difference in death in mitosis cells (%) between 125 nM GSK923295 and 125 nM GSK923295 + 1000 nM navitoclax. #### difference (*p* < 0.0001) in post-mitotic survival cells (%) between 125 nM GSK923295 and 125 nM GSK923295 + 1000 nM navitoclax.

**Figure 4 pharmaceutics-16-00056-f004:**
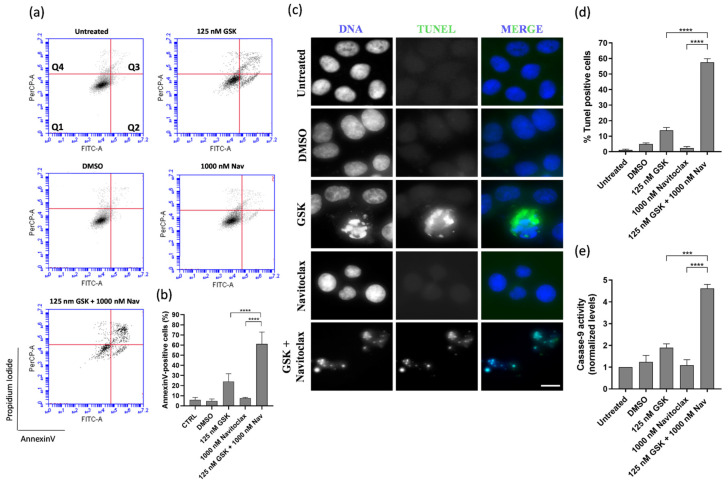
GSK923295 + navitoclax combination enhance A549 lung cancer cell death. Representative cytograms of A549 cell line double stained with Annexin V-FITC and propidium iodide (PI) (**a**). The quadrants Q are defined as Q1 = live (Annexin V- and PI-negative), Q2 = early stage of apoptosis (Annexin V-positive/PI-negative), Q3 = late stage of apoptosis (Annexin V- and PI-positive) and Q4 = necrosis (Annexin V-negative/PI-positive). Quantification of Annexin-V-positive cells (**b**). Representative images of A549 apoptotic cells after 48 h treatment, via TUNEL assay to detect DNA fragmentation (green). DNA (blue) was stained with DAPI. Bar, 5 μm (**c**). Quantification of A549 TUNEL-positive cells (**d**). Quantification of caspase-9 activity was normalized against the protein content of the extract. Additionally, normalization was performed against the value obtained in the untreated group, setting it as 1 for each assay (**e**). Data represent the mean ± SD of three independent experiments, one-way ANOVA followed by Tukey’s multiple comparisons test. *** *p* < 0.001, **** *p* < 0.0001.

**Figure 5 pharmaceutics-16-00056-f005:**
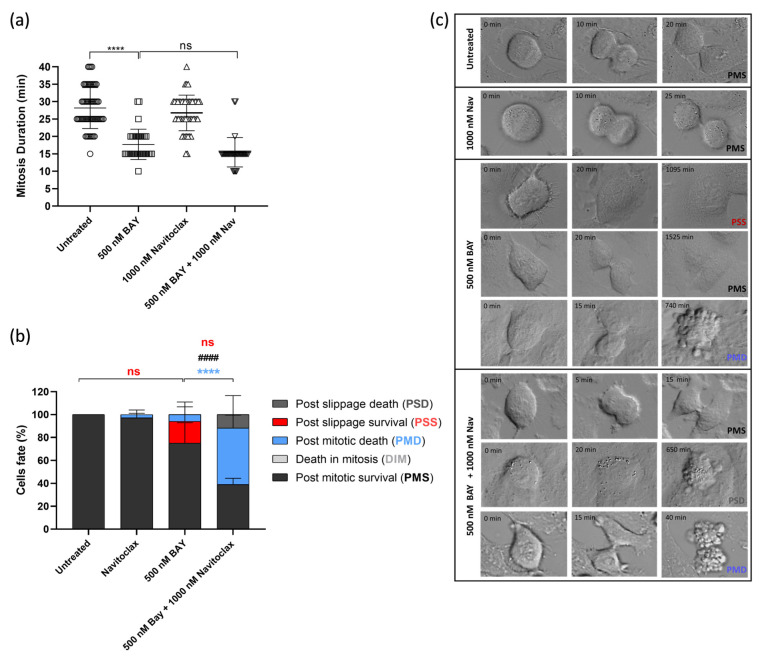
Combination of BAY1217389 + navitoclax induces post-mitotic death in 2D lung cancer cell cultures. Quantification of mitosis duration after respective drug treatments via time-lapse microscopy (**a**). Quantification of cell fate (%) for 48 h using different treatments as indicated (**b**). Representative time-lapse image sequences of A549 cells immediately after drug treatments (**c**). Data represent the mean ± SD of at least three independent experiments, one-way ANOVA followed by Tukey’s multiple comparisons test. **** *p* < 0.0001. ns: not significant difference in mitosis duration between 500 nM BAY1217389 and 500 nM BAY1217389 + 1000 nM navitoclax, and in post-slippage survival cells (%) between untreated and 500 nM BAY1217389 and between 500 nM BAY1217389 and 500 nM BAY1217389 + 1000 nM navitoclax. **** difference in post-mitotic death cells (%) between 500 nM BAY1217389 and 500 nM BAY1217389 + 1000 nM navitoclax. #### difference (*p* < 0.0001) in post-mitotic survival cells (%) between 500 nM BAY1217389 and 500 nM BAY1217389 + 1000 nM navitoclax.

**Figure 6 pharmaceutics-16-00056-f006:**
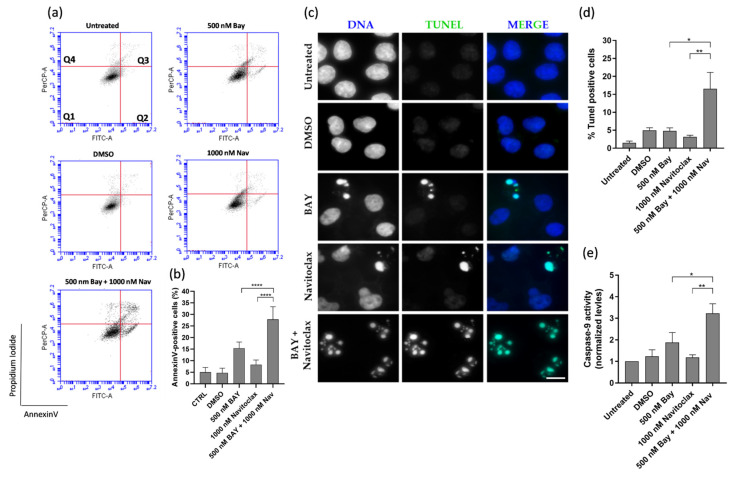
BAY1217389 + navitoclax combination enhances A549 lung cancer cell death by apoptosis. Representative cytograms of A549 cells double stained with Annexin V-FITC and propidium iodide (PI) (**a**). The quadrants Q are defined as Q1 = live (Annexin V- and PI-negative), Q2 = early stage of apoptosis (Annexin V-positive/PI-negative), Q3 = late stage of apoptosis (Annexin V- and PI-positive), and Q4 = necrosis (Annexin V-negative/PI-positive). Quantification of Annexin-V-positive cells (**b**). Representative images of A549 apoptotic cells after 48 h treatment, via TUNEL assay to detect DNA fragmentation (green). DNA (blue) was stained with DAPI. Bar, 5 μm (**c**). Quantification of A549 TUNEL-positive cells (**d**). Quantification of caspase-9 activity was normalized against the protein content of the extract. Additionally, normalization was performed against the value obtained in the untreated group, setting it as 1 for each assay (**e**). Data represent the mean ± SD of three independent experiments, one-way ANOVA followed by Tukey’s multiple comparisons test. * *p* < 0.05; ** *p* < 0.01; **** *p* < 0.0001.

**Figure 7 pharmaceutics-16-00056-f007:**
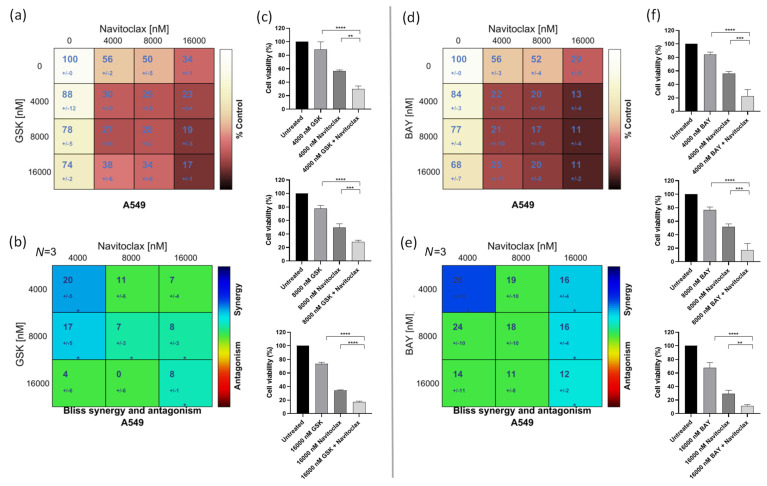
GSK923295 + navitoclax and BAY1217389 + navitoclax combinations enhance cytotoxicity in A549 3D spheroids. Cell viability (%) of single or combination therapies after 48 h of drug exposure (**a**,**d**) from three independent experiments as determined by MTT assay. Synergy scores calculated by the Bliss model of Combenefit software with statistical relevance of * *p* < 0.05. Asterisk indicates synergism effects (**b**,**e**). Three-dimensional spheroid viability (%) after 4000 nM, 8000 nM, and 16,000 nM drug concentration treatments (**c**,**f**). Data represent the mean ± SD of three independent experiments, one-way ANOVA followed by Tukey’s multiple comparisons test. ** *p* < 0.01; *** *p* < 0.001; **** *p* < 0.0001.

**Figure 8 pharmaceutics-16-00056-f008:**
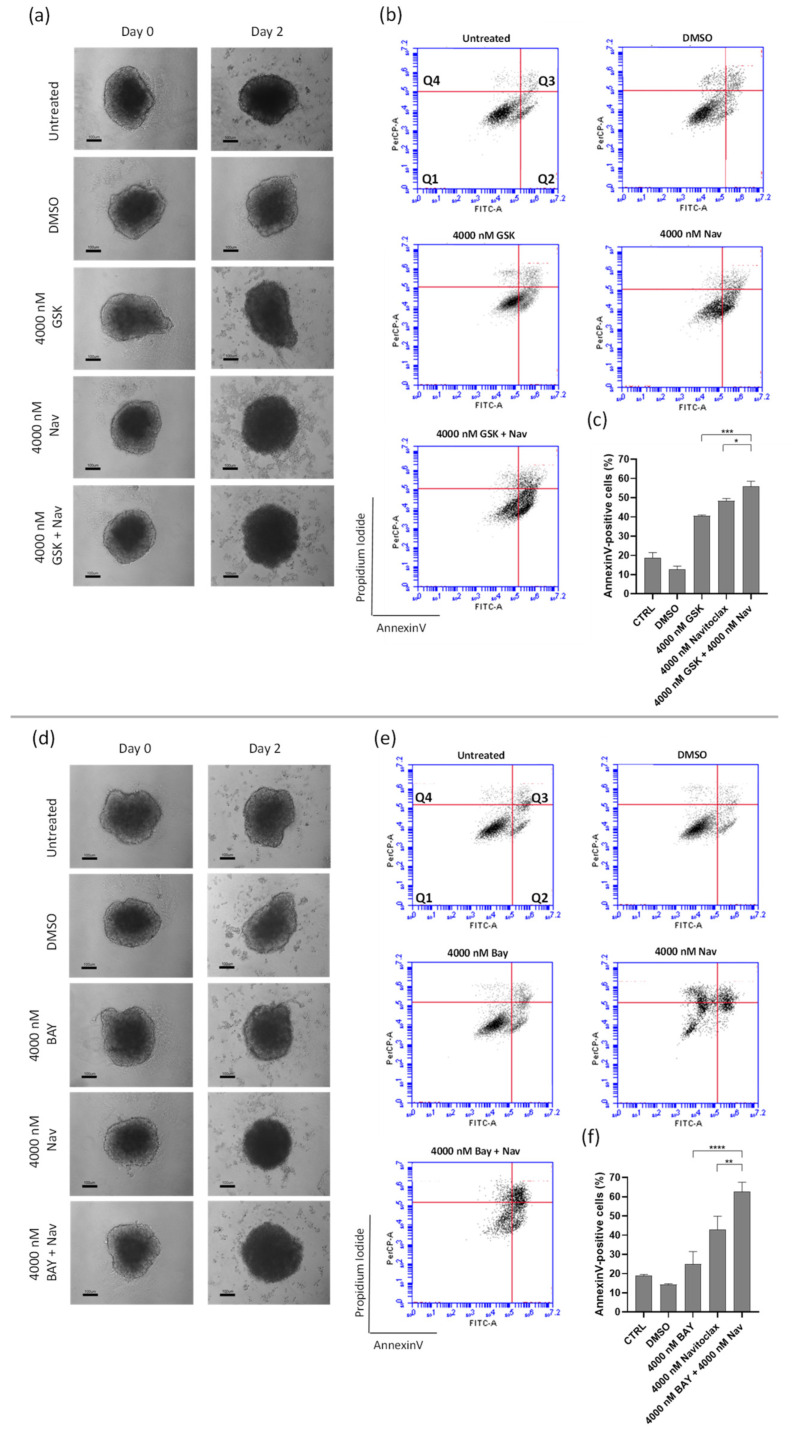
Combination of GSK923295 + navitoclax and BAY1217389 + navitoclax potentiates 3D lung cancer spheroid toxicity. Representative images of A549 3D spheroids at days 0 and 2 post-treatment with mono- or combination drugs (100 μm) (**a**,**d**). Representative cytograms (**b**,**e**) and quantification (**c**,**f**) of Annexin V−positive cells after 48 h of drug exposure. The quadrants Q are defined as Q1 = live (Annexin V− and PI−negative), Q2 = early stage of apoptosis (Annexin V−positive/PI−negative), Q3 = late stage of apoptosis (Annexin V− and PI−positive), and Q4 = necrosis (Annexin V−negative/PI−positive). Data represent the mean ± SD of three independent experiments, one-way ANOVA followed by Tukey’s multiple comparisons test. * *p* < 0.05; ** *p* < 0.01; *** *p* < 0.001 **** *p* < 0.0001.

**Table 1 pharmaceutics-16-00056-t001:** GSK923295 and navitoclax IC_50_ in 2D A549 cells.

Drugs	IC_50_ (nM)
GSK923295	150 ± 30
Navitoclax	13,050 ± 690

**Table 2 pharmaceutics-16-00056-t002:** BAY1217389 and Navitoclax IC_50_ in 2D A549 cells.

Drugs	IC_50_ (nM)
BAY1217389	4340 ± 60
Navitoclax	13,310 ± 910

**Table 3 pharmaceutics-16-00056-t003:** GSK923295 and navitoclax IC_50_ in 3D A549 cells.

Drugs	IC_50_ (nM)
GSK923295	>16,000
navitoclax	6480 ± 1070

**Table 4 pharmaceutics-16-00056-t004:** BAY1217389 and navitoclax IC_50_ in 3D A549 cells.

Drugs	IC_50_ (nM)
BAY1217389	>16,000
navitoclax	6340 ± 930

## Data Availability

The data can be shared upon request.

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
