# Peer review of "Maximizing Anticancer Response with MPS1 and CENPE Inhibition Alongside Apoptosis Induction"

_pharmaceutics, 2023, doi:10.3390/pharmaceutics16010056_

Round 1
Reviewer 1 Report
Comments and Suggestions for Authors
In this manuscript, the authors investigated the potential of combining the BH3 mimetic navito-clax with the mitotic blocker CENPE inhibitor GSK923295 or the mitotic driver MPS-1 inhibitor BAY1217384 in two- and three-dimensional in vitro models of non-small cell lung cancer. The results showed that the two combinations had synergistic cytotoxic activity against two-dimensional cancer cell cultures and induced massive cell death. Overall, this topic is fascinating and provides novel information. However, the manuscript did not thoroughly explore the mechanisms of apoptosis induction. Therefore, I highly recommend accepting the manuscript with minor revisions.
Some specific issues should be addressed as follows:
1. The innovative points of the manuscript should be highlighted in the background section.
2. Some experimental results need to provide quantitative data, such as colony formation experiments.
3. The scale of many of the figures is too small to be seen clearly.
4. Is it possible to investigate changes in the expression of key factors during mitosis through WB or qPCR experiments?
5. The mechanism of apoptosis should be explored in depth.
6. An additional conclusion section should be included to summarize the findings of the manuscript and provide an outlook.
7. The author must thoroughly check the manuscript for any other mistakes.
Comments on the Quality of English LanguageThe manuscript should be carefully checked and revised to avoid the spelling, expression and grammar errors.
Reviewer 2 Report
Comments and Suggestions for Authors
In the manuscript by Bárbara Pinto et al. entitled “ Maximizing Anticancer Response with MPS-1 and CENPE Inhibition Alongside Apoptosis Induction”; explores the synergistic cytotoxic activity of combining the BH3-mimetic navitoclax with either the CENPE inhibitor GSK923295 or the MPS-1 inhibitor BAY1217384 in 2D and 3D in vitro models of non-small cell lung cancer (NSCLC). The results are well-documented, and the discussion provides valuable insights into the observed effects. One commendable aspect of the study is the comprehensive evaluation of the combinations in both 2D and 3D models, shedding light on potential differences in drug response influenced by factors such as drug penetration, microenvironment, and cellular interactions. The authors appropriately acknowledge the need for further investigation, particularly in the clinical context, given the observed variations between the two combinations in 3D spheroids.
Based on my assessment, with further minor adjustments, the manuscript could be considered for publication.
General comments:
Given the authors' emphasis on antimitotic treatments in lung cancer, I could suggest starting the introduction section with "Microtubule-targeting agents (MTA) such as paclitaxel (Line 43) and then introducing lung cancer.
In the Introduction, you might consider breaking the long sentence starting with "Microtubule-targeting agents (MTAs)..." into smaller sentences for better readability.
Ensure consistent use of terminology: "BH3-mimetic" and "BH3 mimetic" (use one form consistently throughout).
In tables, consider correcting IC50 to IC50.
In the Discussion, when discussing the discrepancies between 2D and 3D models, consider elaborating on potential reasons for the observed differences. Are there specific characteristics of the 3D model that could explain the variations in drug sensitivity?
In the Methods section, clarify the significance of using specific concentrations for inhibitors. Is there prior literature supporting the chosen concentration ranges?
Confirm that there is a logical and smooth flow between sections, ensuring that each subsection naturally leads to the next.
Consider breaking down the discussion by adding a conclusion section for the last paragraph.
Correct the reference 27 (doi…).
Reviewer 3 Report
Comments and Suggestions for Authors
This is an interesting investigation that describes how to improve the antiproliferative response of MPS1 and CENPE inhibitors when used in combination with Navitoclax.
Comments:
1) Refer to Figure 1b. CENPE bands were not seen (very faint) on the western blot in NCI-H460. Because two gels (provided in supplentary) have more CENPE in HPAEpic and two blots have less, the results cannot be confirmed. Is it possible to repeat it with the maximum amount of exposure? Because the bands are so faint, use a more concentrated primary antibody or a more concentrated secondary antibody (1:1000 or less dilution). Also, use 8%-12% gel and transfer for a longer period so that greater molecular weight proteins may transfer more effectively from gel to blot.
2)The authors published an article titled "Navitoclax Enhances the Therapeutic Effects of PLK1 Targeting on Lung Cancer Cells in 2D and 3D Culture Systems" recently. There, the authors used the inhibitor that targets the spindle assembly checkpoint same like in this manuscript. So, what is the significance of this work, and how effective is targeting MPS1 or CENPE in combination with Navitoclax versus PLK1 alone? You might have simply integrated all of these data to obtain more supportive evidence demonstrating how successfully we can improve microtubule targeting inhibitors through combination treatment.
3) The authors of this publication are attempting to improve the cytotoxic impact of the inhibitors GSK923295 and BAY1217384. However, when it comes to 3D culture. Navitoclax began to exhibit more promising results than the other inhibitors (more than 2.5 times, because the IC50 of other inhibitors are yet identify). Which indicates that when it comes to tumor 3D or in vivo, other inhibitors, are ineffective. What is the use of using those inhibitors in tumor 3D or future in vivo studies if they have little effect? We may not be able to say that Navitoclax improves the antiproliferative properties of GSK923295 and BAY1217384 in this experiment. In fact, you showed that how to improve the impact of Navitoclax by utilizing nontoxic inhibitors like GSK923295 and BAY1217384 in 3D (Figure 7). Therefore your 2D and 3D data are quite opposite and difficult to conclude.
4) Figure 7 table 3 title, it written 2D instead of 3D. Please correct the typo.
